# Identifying the Active Phase of $RuO_2$ in the Catalytic CO Oxidation Reaction, Employing Operando CO Infrared Spectroscopy and Online Mass Spectrometry

**Phillip Timmer [1,2], Lorena Glatthaar [1,2], Tim Weber [1,2] and Herbert Over [1,2,*]**

[1] Institute of Physical Chemistry, Justus Liebig University, Heinrich Buff Ring 17, 35392 Giessen, Germany; phillip.timmer@phys.chemie.uni-giessen.de (P.T.); lorena.glatthaar@phys.chemie.uni-giessen.de (L.G.)

[2] Center for Materials Research, Justus Liebig University, Heinrich-Buff-Ring 16, 35392 Giessen, Germany

[*] Correspondence: herbert.over@phys.chemie.uni-giessen.de

**Abstract:** Operando diffuse reflectance infrared Fourier transform spectroscopy (DRIFTS) is combined with online mass spectrometry (MS) to help to resolve a long-standing debate concerning the active phase of $RuO_2$ supported on rutile $TiO_2$ ($RuO_2@TiO_2$) during the CO oxidation reaction. DRIFTS has been demonstrated to serve as a versatile probe molecule to elucidate the active phase of $RuO_2@TiO_2$ under various reaction conditions. Fully oxidized and fully reduced catalysts serve to provide reference DRIFT spectra, based on which the operando CO spectra acquired during CO oxidation under various reaction conditions are interpreted. Partially reduced $RuO_2@TiO_2$ was identified as the most active catalyst in the CO oxidation reaction. This is independent of the reaction conditions being reducing or oxidizing and whether the starting catalyst is the fully oxidized $RuO_2@TiO_2$ or the partially reduced $RuO_2@TiO_2$.

**Keywords:** catalytic CO oxidation; ruthenium; $RuO_2$; catalytically active phase; operando DRIFTS





## 1. Introduction

Scientific discussions about the nature of the active phase in a catalytic reaction are not straightforward [1]. This has been encountered particularly often with reducible oxides of precious metals such as Ru [2,3], Pd [4,5], Pt [6], Rh [7], and Ir [8–10], which can readily adapt their oxidation state depending on the specific reaction condition. Therefore, to identify the actual active phase one needs to employ operando spectroscopic or structure-sensitive methods [11–13].

One particular intensively discussed example, and the first catalytic system where this discussion heated up, is the CO oxidation reaction over Ru-based catalysts [14]. Here, two schools are involved: one that prefers metallic ruthenium being the active phase [15] and another that favors oxide being the active phase [16]. We recall that the ruthenium system reveals a surprisingly rich chemistry during CO oxidation, exhibiting phase changes and being subject to poisoning by the formed $CO_2$ [2] to the point that even oscillations in the $CO_2$ yield can occur in the CO oxidation reaction performed in a flow cell [17]. At the summit of this discussion, ruthenium dioxide was even considered to be not active at all in oxidation catalysis. Admittedly, the catalytic CO oxidation reaction over Ru-based catalysts does not have application in exhaust after treatment due to the potential formation of toxic and volatile $RuO_4$ at high temperatures. However, Ru and especially $RuO_2$ are currently applied in large-scale industrial processes, such as the catalytic HCl oxidation reaction (Deacon process) [18–20] and the electrochemical chlorine evolution reaction (CER) [21], and is considered to be the most efficient oxygen evolution reaction (OER) catalyst for electrochemical water splitting under acidic reaction conditions [22,23]. A general discussion about the catalytically active phase even for a "seemingly less relevant" CO oxidation reaction may therefore have greater impact than hitherto expected, since the

same sites and phases may play a role in these reactions as well. As far as we can judge, the discussion of the active phase of Ru-catalyzed CO oxidation has still not been settled.

In a recent paper by Gustafson et al. [7], the discussion of the active phase in the CO oxidation over Pd and Rh was settled, employing operando high-pressure X-ray photoelectron spectroscopy (XPS). For Rh, the oxygen-covered metallic surface was shown to be more active than the oxide, whereas for Pd, thin oxide films were reported to be at least as active as the metallic surface, but a thicker oxide was less active.

In this contribution, we present and discuss operando diffuse reflectance infrared Fourier transform spectroscopy (DRIFTS) experiments [24,25] in combination with online mass spectrometry for catalytic CO oxidation over ruthenium. Catalytic CO oxidation, a well-documented model reaction [26], is carried out in a flow cell reactor setup under various reaction conditions. DRIFTS has been demonstrated to be a powerful technique to identify reaction intermediates on the catalyst's surface, in particular when CO is involved. More important for our present study is, however, that CO can serve as a versatile probe molecule to study the actual chemical nature of the active phase under reaction conditions [27,28]. This approach is applied to elucidate the active phase of $RuO_2@TiO_2$. To do so, first, fully oxidized and fully reduced Ru-based catalysts supported on rutile $TiO_2$ are prepared. The DRIFT spectra of these are used as reference spectra for the subsequent interpretation of operando CO spectra acquired during CO oxidation under various reaction conditions. It is found that, independent of the reaction conditions, the partially reduced $RuO_2@TiO_2$ catalyst constitutes the most active catalyst in the CO oxidation reaction.

## 2. Experimental Results

### 2.1. Characterization of Pre-Oxidized and Pre-Reduced $RuO_2@TiO_2$ and $Ru^0$ + $TiO_2$ Samples

Figure 1 shows XP spectra of the Ru 3d binding energy region of supported $RuO_2$ on $TiO_2$, referred to as $RuO_2@TiO_2$, and the metallic $Ru^0$ physically mixed with $TiO_2$, referred to as $Ru^0$ + $TiO_2$. The fit parameters used were taken from Morgan et al. [29] and are compiled in Table S1. According to the spectra, ruthenium was fully oxidized in the case of $RuO_2@TiO_2$. The spectra of the mixture of $Ru^0$ + $TiO_2$ yielded very low signals in XPS. Therefore, pure $Ru^0$ powder was used to make a meaningful deconvolution possible in the XPS analysis. Here, pure metallic $Ru^0$ was found. After exposure to reducing CO oxidation conditions (1% $O_2$/4% CO/95% Ar), the XP spectra of $Ru^0$ + $TiO_2$ and $RuO_2@TiO_2$ in Figure 1 indicate spectral features of both metallic and oxidic Ru. For $Ru^0$ + $TiO_2$, this effect seem to be less pronounced than for $RuO_2@TiO_2$. The reason for this is likely the large size of the $Ru^0$ particles, as only the surface would be oxidized and therefore the bulk metal signal would dominate the XPS signal.

### 2.2. CO DRIFTS Experiments of Oxidized and Reduced $RuO_2@TiO_2$ and $Ru^0$ + $TiO_2$ Samples

Figure 2A shows CO DRIFT spectra of $RuO_2@TiO_2$ and $Ru^0$ + $TiO_2$. For this and all following DRIFTS spectra, the y-axis corresponds to absorption in arbitrary units. As evidenced by XPS, these samples consisted of pure oxide and pure metal, respectively. As such, these served as references for assigning spectral DRIFTS features to the CO adsorption on oxide $RuO_2$ and pure metal $Ru^0$, respectively. For both samples, a single symmetric band at around 2060 $cm^{-1}$ was observed. The same spectral feature was observed for $Ru^0$ + $TiO_2$, which had been oxidized in 4% $O_2$ at 300 °C (cf. Figure S2). We therefore conclude that a distinction between pure metal and pure oxide cannot be made in DRIFTS experiments based on band position alone.

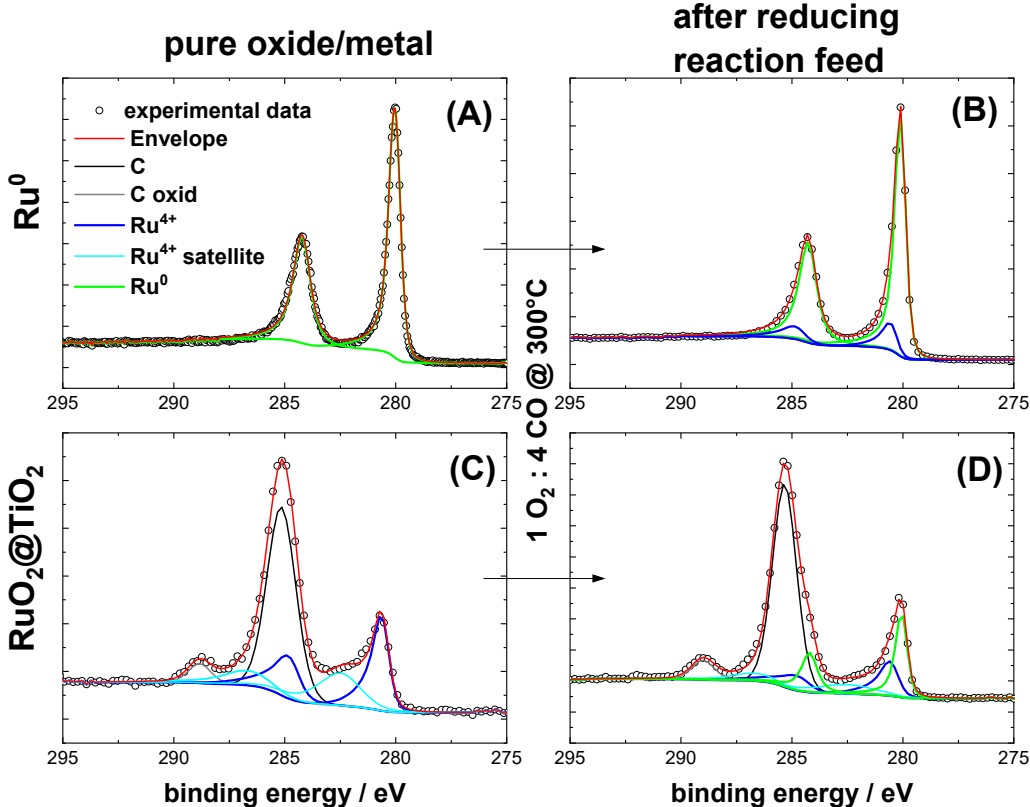

**Figure 1.** Ru 3d XP spectra of (**A**) Ru⁰ and (**C**) RuO₂@TiO₂. The former sample is measured without TiO₂, as the signal is too weak otherwise. (**B**,**D**) show the respective samples after exposure to reducing CO oxidation conditions (1% $O_2$/4% CO/95% Ar) at 300 °C. Table S1 provides the fit parameters for deconvolution of the experimental spectra (circles). The fit parameters are taken from Morgan et al. [29].

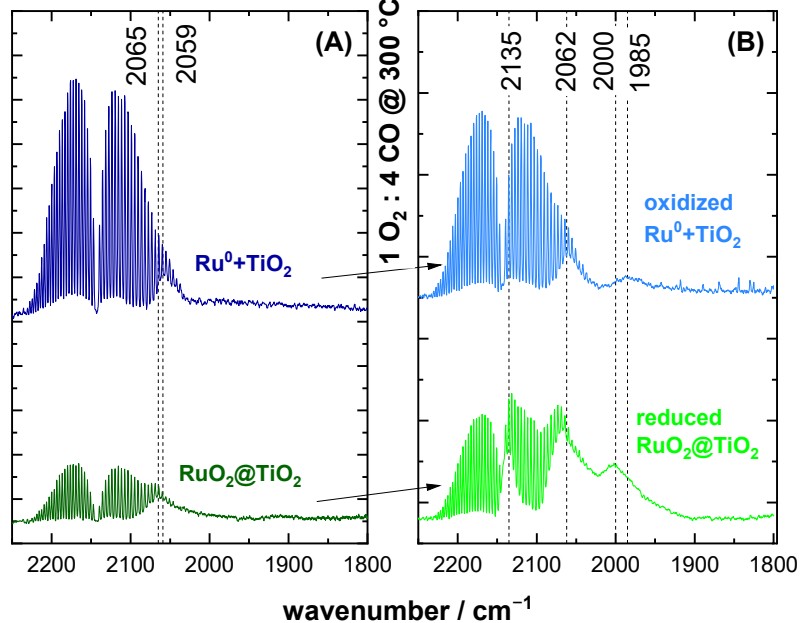

**Figure 2.** DRIFT spectra of CO adsorption are shown for (**A**) commercial Ru⁰ + TiO₂ containing only purely metallic Ru and RuO₂@TiO₂ containing only purely oxidic Ru as prepared by Pechini synthesis on the left. (**B**) Samples after exposure to reducing CO oxidation conditions (1% $O_2$/4% CO/95% Ar) at 300 °C. The spectra are recorded at room temperature.

Under reducing CO oxidation conditions (1% $O_2$/4% CO/95% Ar) at 300 °C, the as-prepared catalyst changed its composition in that $Ru^0$ + $TiO_2$ partially oxidized, whereas $RuO_2$@$TiO_2$ partially reduced. DRIFT spectra of the samples after cooling down under these reaction conditions are summarized in Figure 2B. Here, some differences and similarities of the two samples can be identified. The differences in signal strength were due to differing reflectivity of the samples. For oxidized $Ru^0$ + $TiO_2$ the only difference to the pristine $Ru^0$ + $TiO_2$ was the occurrence of a second absorption band at 1985 $cm^{-1}$. For $RuO_2$@$TiO_2$, the spectra changed more profoundly relative to its pristine sample (Figure 2A). On the one hand, it also exhibited the aforementioned second signal, albeit at a higher wavenumber of 2000 $cm^{-1}$. In addition, the band at 2065 $cm^{-1}$ revealed an asymmetric shoulder reaching lower wavenumbers. Experiments on more strongly reduced $RuO_2$@$TiO_2$ indicated this shoulder to be a distinct third species at ca. 2040 $cm^{-1}$ (cf. Figure 3). The importance of this species for the activity of $Ru^0$/$RuO_2$ towards CO oxidation is discussed below. Furthermore, a fourth band at 2135 $cm^{-1}$ can be observed. Spectra of pure $TiO_2$ under a CO atmosphere did not indicate any adsorbed CO in DRIFTS (cf. Figure S3). The DRIFT spectra of $RuO_2$@$TiO_2$ and $Ru^0$ + $TiO_2$ in Figure 2B under reducing reaction conditions were similar but different from the pristine samples, thus evidencing a partial reduction of $RuO_2$@$TiO_2$ and the partial oxidation of $Ru^0$ + $TiO_2$ in Figure 2B.

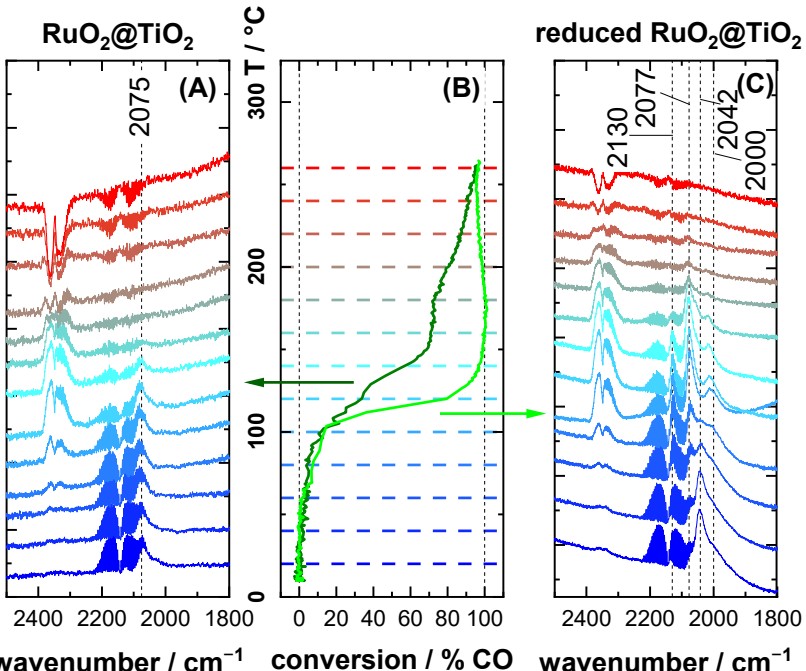

**Figure 3.** Operando DRIFT spectra of (**A**) $RuO_2$@$TiO_2$ and (**C**) reduced $RuO_2$@$TiO_2$ as well as (**B**) corresponding CO conversion data under oxidizing (2% $O_2$/2% CO/96% Ar) reaction feed composition. For the conversion data, the dark green line corresponds to $RuO_2$@$TiO_2$ and the light green line to reduced $RuO_2$@$TiO_2$. The temperature axis of the conversion plot is marked by dashed lines in the color of the corresponding DRIFT spectra. Spectra are recorded in 20 °C increments. The heating ramp is 1.8 K·$min^{-1}$.

### 2.3. CO Oxidation Experiments of RuO_2@TiO_2 Samples

Figure 3 shows DRIFT spectra and the corresponding CO conversion of $RuO_2$@$TiO_2$ and reduced $RuO_2$@$TiO_2$ (see Experimental Details, Section 4.1) during the CO oxidation reaction under oxidizing conditions (2% $O_2$/2% CO/96% Ar) in the temperature range of 20 °C to 260 °C.

For $RuO_2$@$TiO_2$, presented in Figure 3A, a single band at 2075 $cm^{-1}$ was observed, in accordance with a fully oxidized $RuO_2$ surface (cf. Figure 2A). This band remained unchanged in shape and position up to 140 °C, where it started to diminish. Above 160 °C,

no adsorbed CO could be detected in DRIFTS. The conversion increased till it reached a plateau from 150 °C to 180 °C, at which temperature the reaction rate increased again up to 260 °C.

The reduced $RuO_2$@$TiO_2$, which was exposed to 4% CO at 300 °C during the pretreatment, initially showed, as seen in Figure 3C, a somewhat different peak shape than in Figure 2B. Here the most prominent feature is a band at 2042 cm$^{-1}$ surrounded by shoulders reaching high and low wavenumbers. At 60 °C, the 2042 cm$^{-1}$ signal started to diminish, and at 100 °C it vanished completely. Between 120 °C and 180 °C, the DRIFT spectra looked like those of the reduced $RuO_2$@$TiO_2$ sample (cf. Figure 2B), albeit with a less pronounced low wavenumber shoulder on the 2077 cm$^{-1}$ band. During this transition, it can clearly be seen that the DRIFT bands of adsorbed CO observed between 2100 cm$^{-1}$ and 1980 cm$^{-1}$ consisted of three distinct spectral features at ca. 2075 cm$^{-1}$, 2000 cm$^{-1}$, and 2042 cm$^{-1}$. Although the 2042 cm$^{-1}$ feature vanished first with increasing temperature, the bands at 2075 cm$^{-1}$ and 2000 cm$^{-1}$ seemed to be more stable. It is furthermore important to note that, concomitant with the disappearance of the 2042 cm$^{-1}$ band at 100 °C, the reaction rate increased steeply. Above 100 °C, the conversion observed for reduced $RuO_2$@$TiO_2$ overtook the one for $RuO_2$@$TiO_2$ and remained higher than the one for $RuO_2$@$TiO_2$ over the entire temperature range. At high temperatures, the gas-phase bands of CO and $CO_2$ seemed to become negative. This is due to IR emission of the heated gas layer above the catalyst.

Figure 4 depicts DRIFT spectra of $RuO_2$@$TiO_2$ during cooling from 260 °C (cf. Figure 3A) to room temperature under an oxidizing (2% $O_2$/2% CO/96% Ar) reaction feed. Interestingly, below 160 °C the spectra showed the same low wavenumber bands previously associated with partial reduction of $RuO_2$@$TiO_2$ (cf. Figure 2B), albeit to a lesser degree. This suggests that a partial reduction of the $RuO_2$ surface is observed, even under an oxidizing reaction feed. Further implications of this finding are discussed in the Section 3.

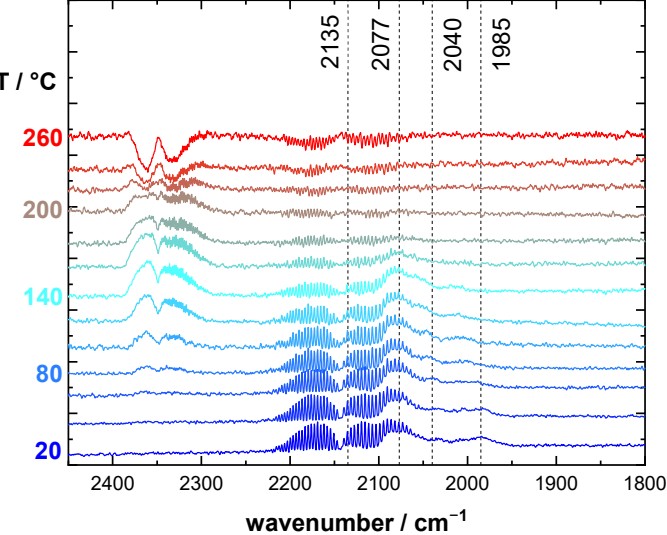

**Figure 4.** Operando DRIFT spectra of $RuO_2$@$TiO_2$ during cooldown under oxidizing (2% $O_2$/2% CO/96% Ar) reaction feed composition. Spectra are recorded in 20 °C increments. The heating ramp is 1.8 K·min$^{-1}$.

Figure 5 summarizes the operando DRIFTS experiments of $RuO_2$@$TiO_2$ and reduced $RuO_2$@$TiO_2$ during the CO oxidation reaction under reducing conditions (1% $O_2$/4% CO/95% Ar) when increasing the reaction temperature from 20 °C to 260 °C together with corresponding CO conversion data. In DRIFTS of $RuO_2$@$TiO_2$ (Figure 5A), there was again only a single band at 2077 cm$^{-1}$ observed at low temperatures. This spectral feature remained unchanged up to 140 °C, when two additional bands appeared at 2025 cm$^{-1}$ and 2134 cm$^{-1}$. The conversion under reducing reaction conditions (Figure 5B) behaved

like that under oxidizing reaction conditions (Figure 3B), with conversion increasing up to 140 °C, followed by a plateau till 210 °C. At this temperature, the band at 2040 cm$^{-1}$ re-emerged, concomitant with a steep increase in the conversion.

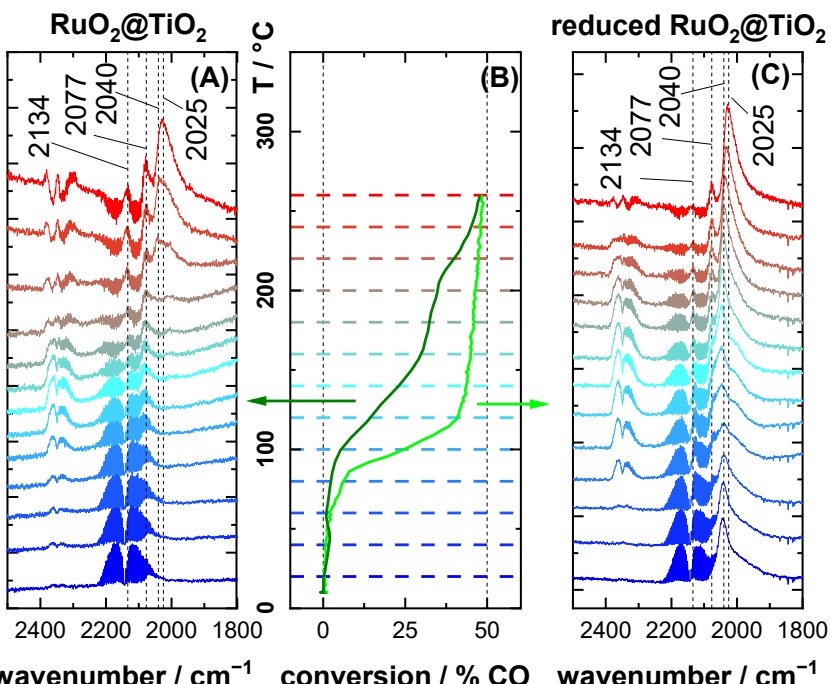

**Figure 5.** Operando DRIFT spectra of (**A**) RuO$_2$@TiO$_2$ and (**C**) reduced RuO$_2$@TiO$_2$, as well as (**B**) corresponding CO conversion data under reducing (1% O$_2$/4% CO/95% Ar) reaction feed composition. For the conversion data, the dark green line corresponds to RuO$_2$@TiO$_2$ and the light green line to reduced RuO$_2$@TiO$_2$. The temperature axis of the conversion plot is marked by dashed lines in the color of the corresponding DRIFT spectra. Spectra are recorded in 20 °C increments. The heating ramp is 1.8 K·min$^{-1}$.

With a further increase in the temperature, the 2040 cm$^{-1}$ band diminished and merged with the 2025 cm$^{-1}$ signal. At 260 °C, only bands at 2134 cm$^{-1}$, 2075 cm$^{-1}$, and 2025 cm$^{-1}$ could be discerned clearly, with the latter being the most prominent one. Different from RuO$_2$@TiO$_2$ under oxidizing conditions, here a clear correlation between the partial reduction of the catalyst and an increase in activity was observed.

For DRIFTS of reduced RuO$_2$@TiO$_2$ under reducing reaction conditions (Figure 5C), the bands of adsorbed CO initially looked like those observed under oxidizing reaction conditions (Figure 3C). However, with increasing temperature, the high and low wavenumber shoulders of the 2040 cm$^{-1}$ signal became more pronounced. Here, the 2040 cm$^{-1}$ did not vanish above 100 °C. Instead, it remained clearly visible up to 200 °C, where it started to diminish. Note that for RuO$_2$@TiO$_2$ and reduced RuO$_2$@TiO$_2$ the spectra at 260 °C started to look very similar. The CO conversion of reduced RuO$_2$@TiO$_2$ under reducing conditions was markedly higher than that of RuO$_2$@TiO$_2$ throughout the entire temperature region.

Conversion plots and DRIFT spectra for a second heating ramp are summarized in Figure 6. The spectra were recorded after the catalyst was cooled back to room temperature. For both reducing and oxidizing reaction conditions, the DRIFT spectra looked similar regardless of the initial state of the catalyst. For reducing conditions, both samples exhibited bands at ca. 2135 cm$^{-1}$, 2060 cm$^{-1}$, and 2000 cm$^{-1}$, with almost identical shape and intensity (cf. Figure 6B, green spectra), which were characteristic for the partial reduction of the catalyst. For oxidizing conditions, the main band was 2077 cm$^{-1}$ (cf. Figure 6B, blue spectra), but additionally, some weak signals were observed reaching down to 2000 cm$^{-1}$, as also seen in Figure 4. The conversion curves converged as well for the second heating ramp for all samples only depending on the gas feed composition. For reducing conditions,

the conversion curves (Figure 6A) were practically identical. The only difference with respect to the first heating ramp of reduced RuO₂@TiO₂ (shown in grey) is that the conversion was slightly lower throughout the temperature range and the conversion during the second heat-up of RuO₂@TiO₂ still exhibited hints of the high temperature conversion plateau.

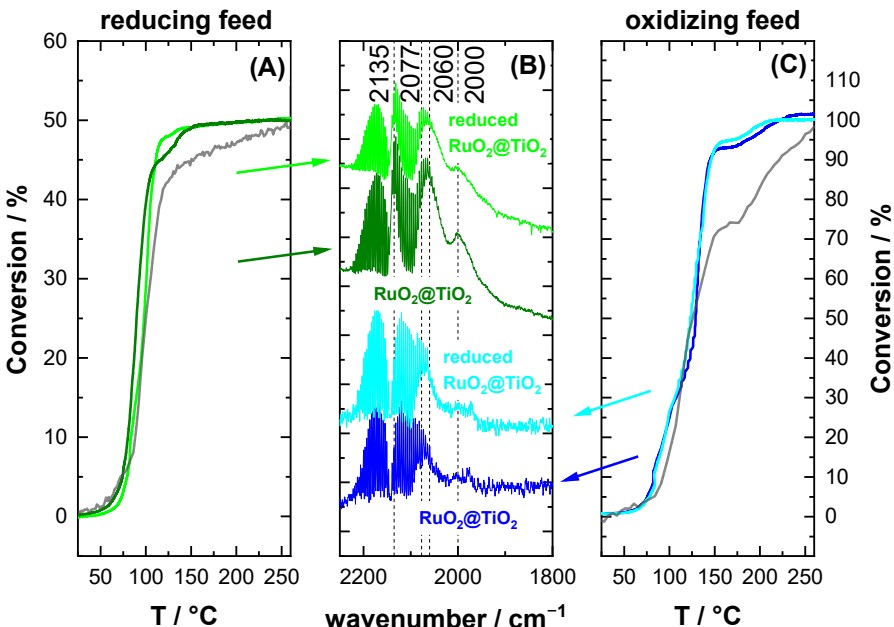

**Figure 6.** Operando DRIFT spectra and conversion of RuO₂@TiO₂ and reduced RuO₂@TiO₂ under (**A**) reducing and (**C**) oxidizing reaction feed composition for the second reaction heat-up. The spectra (**B**) are recorded after cooling the catalyst to room temperature. The grey conversion curves represent the conversion of the first heat-up for the RuO₂@TiO₂ and reduced RuO₂@TiO₂, as shown in Figures 3 and 5, respectively. Regardless of the initial state of the catalyst, spectra and conversion converged in accordance with the reaction feed.

For oxidizing conditions (Figure 6C), the differences in the conversion curves were more profound between the first and second heat ramps. The conversion plateau of the second heating ramp was markedly reduced relative to the first heat-up, signifying a clear correlation between activity and the presence of low wavenumber signals. However, the conversion curves for the second heat-up for both RuO₂@TiO₂ and reduced RuO₂@TiO₂ were practically identical and depended only on the reaction conditions. Overall, we can conclude from Figure 6 that the catalyst adapted dynamically to the same active phase regardless of whether it started from the oxidized or the reduced sample, but of course depending on the reaction environment.

## 3. Discussion

### 3.1. Stretching Vibrations of Adsorbed CO Probing the Actual Surface Oxidation State of RuO₂

Table 1 summarizes stretching modes of adsorbed CO for various Ru-based catalysts reported in the literature. We note that the assignment of the band positions to specific adsorption sites vary sometimes between different publications. This is likely due to exact band positions being dependent on many factors, such as coverage of CO and O, as well as the oxidation state of the adsorption site. A few general trends can, however, be identified. RuO₂ bands below 2000 cm⁻¹ are assigned to CO on bridge positions, whereas those above 2000 cm⁻¹ are ascribed to on-top CO. Here, a bridge position means CO being adsorbed to two adjacent Ru atoms, whereas on-top means CO on a single Ru atom. Infrared bands between 2000 and 2050 cm⁻¹ were mainly observed for reduced RuO₂ or supposedly metallic samples. Note that Peden et al. [30], who assigned the band at 2040 cm⁻¹ to metallic Ru, exposed the sample to strongly oxidizing conditions before the measurement.

Bands in the region of 2050–2100 cm$^{-1}$ are assigned to both $RuO_2$ and metallic Ru. Bands above 2100 cm$^{-1}$ are assigned to $Ru^{x+}(CO)_y$, which is linked to partial reduction of $RuO_2$ or to CO on fully O covered $RuO_2(110)$ surfaces in some single-crystal studies.

**Table 1.** Vibrational band positions of CO on a Ru-based catalyst, as reported in the literature.

| Band Position/cm$^{-1}$ | Adsorbed Species | Substrate | Ref. | Remarks |
|---|---|---|---|---|
| 1860–1970 | CO | Mildly reduced $RuO_2(110)$ | [31] | Bridging CO being the majority species |
| 2070–2080 | CO | Mildly reduced $RuO_2(110)$ | [31] | On-top and bridging CO couple to a single band |
| 2000–2050 | [$Ru^0$-CO]-linear | $Ru/Al_2O_3$ | [32] | Terminal CO |
| 2010–2070 | terminal CO | Ru-carbonyl | [33,34] | Depends on the carbonyl size |
| 2048 | CO | Ru(001) | [30] | Measured after exposure to oxidizing conditions; likely partially oxidized |
| 2068 | $(CO+O)(2 \times 2)$ | Ru(001) | [35] | -- |
| 2080 | CO | Ru(001) | [30] | Measured under strongly oxidizing conditions at 500 K, likely oxidized |
| 2100–2123 | CO | Stoichiometric $RuO_2(110)$ | [31] | On-top CO on stoichiometric $RuO_2(110)$ |
| 2100–2135 | CO/O | $RuO_2(110)$ | [36] | On-top CO being the majority species |
| 2100–2150 | CO/O | $RuO_2(110)$ | [31] | O being the majority species; on-top CO embedded in O matrix |
| 2070, 2130 | $Ru^{x+}$-$(CO)_3$ | $Ru/Al_2O_3$ | [32,37] | x = 1–3 |
| 2134 | $Ru^{x+}(CO)_y$ | $Ru/SiO_2$ | [38] | Linked to oxidation of Ru |
| 2135 | $Ru^{x+}(CO)_y$ | Reduced $RuO_2(110)$ | [3] | x and y undetermined; linked to surface roughening due to reduction |

The CO adsorption on $RuO_2@TiO_2$ and $Ru^0 + TiO_2$ at 20 °C resulted in one distinct DRIFT signal at around 2060 cm$^{-1}$. Although the band position was slightly different between the two catalysts, it varied more substantially due to coverage effects and the occupancy of bridge positions by oxygen. Accordingly, this spectral feature alone was insufficient for characterization of the chemical state of the catalyst. It is quite surprising that only one single band was observed, as there are many different facets and adsorption sites expected to be present on the particle surfaces. The presence of only one symmetrical band in DRIFTS points toward efficient dipole–dipole coupling of the vibrational modes of all the different sites next to each other. This coupling was previously reported for RAIRS of CO on mildly reduced $RuO_2(110)$ [31].

When $RuO_2@TiO_2$ and $Ru^0 + TiO_2$ were exposed to a reducing reaction feed (1% $O_2$/4% CO/95% Ar) at 300 °C, the samples were partially reduced and partially oxidized, respectively, as revealed in XP spectra (cf. Figure 1). This state of partial reduction of $RuO_2@TiO_2$ or partial oxidation of $Ru^0 + TiO_2$ was also corroborated by a dedicated DRIFTS signal at ca. 2000 cm$^{-1}$ for both samples (cf. Figure 2). Additionally, a mid-wavenumber signal around 2040 cm$^{-1}$ appeared for the reduced $RuO_2@TiO_2$ catalyst. This species was only visible as a shoulder in Figure 2B but, however, turned into an individual band when $RuO_2$ was reduced with 4% CO, as seen in the DRIFT spectra recorded at 20 °C in Figures 3C and 5C. Furthermore, a DRIFTS signal at 2135 cm$^{-1}$ was discerned for reduced $RuO_2@TiO_2$, which can likely be attributed to a Ru-carbonyl species due to reduction induced roughening of the catalyst.

### 3.2. CO Oxidation as Case Study

The active phase of Ru-based catalysts in the CO oxidation reaction has been controversially debated over the last two decades. Broadly speaking, there are three interpretations of what constitutes the most active phase: (1) a metal surface with chemisorbed O reacting

with CO without oxidizing the Ru itself [15,39–41], (2) an oxide surface with CO binding to coordinatively unsaturated Ru sites ($Ru_{cus}$) and reacting with adjacent bridging oxygen [31,36,42–44], and (3) a sub-stoichiometric $RuO_x$ or mixture of phases. We elaborate on these different views on the active phase in the following sections.

Various UHV studies reported either metal or oxide to be the most active phase. On the one hand, Goodman and coworkers claimed to have identified metallic Ru as the active phase. This determination conflicts, however, with studies demonstrating that at low O coverages oxygen binds too strongly and at high O coverages CO binds too weakly [3,16,45], rendering metallic Ru inactive according to the Sabatier principle. This view is supported by the experiments of Narloch et al. [46], wherein CO desorbed from a mixed CO-O phase on Ru(0001) without forming $CO_2$. In addition, it is important to note that in the study of Gao et al. [3], no structural information of the active phase was provided except for post-reaction Auger electron spectroscopy (AES) (coverage of oxygen was found to be close to one monolayer). We note that AES characterization was carried out after heating the sample to desorb residual CO from the surface. This procedure may have reduced any potentially present surface oxide.

On the other hand, $RuO_2$ has been favored as the active phase in CO oxidation by various other studies [31,36,42–44]. Gao et al. argued that $RuO_2$ is not active in catalytic CO oxidation due to its high adsorption energy for CO, leading to poisoning by CO [3] and an expected reaction order of –1 in CO. Meanwhile, this conclusion was disproven by Martynova et al. [47], who determined the reaction orders for CO oxidation over $RuO_2$(110) to be +1 for CO and zero for $O_2$. A reaction order of +1 in CO is compatible with a previous study of Seitsonen et al. [48], who found not only strongly but also weakly adsorbed CO on $RuO_2$(110).

So far, we have considered results for single-crystal surfaces without considering defects like steps or edges. On powder catalysts, the abundance of steps, edges, and corners may provide sites with more favorable adsorption energies for both O and CO, as demonstrated by Kim et al. [49] and Šljivančanin et al. [50]. The importance of defects for Ru-catalyzed CO oxidation has already been discussed by Gao et al.: Defects may overcome unfavorable adsorption energies of Ru(0001) [3].

Let us now discuss the third option of multiple phases in coexistence. One motive of such a multi-phase system, discussed in the literature and often linked to increased activity, consists of an ultrathin layer of $RuO_2$ or sub-stoichiometric $RuO_x$ over Ru-metal [16,47,51–53]. Martynova et al. [47] demonstrated that the activity of Ru(0001) increased substantially when a surface oxide layer of 1–7 ML grew and that its activity was even higher when this surface oxide was disordered.

Another motive discussed in the literature is that of oxide and reduced oxides or even metallic phases coexisting on the surface [42,47,54,55]. Blume et al. [55] identified with XPS microscopy that oxidized $RuO_2$ and reduced $RuO_x$ areas coexist on Ru(0001) during CO oxidation. They correlated the coexistence of both phases with increased activity. Martynova et al. [47] demonstrated that Ru(0001) formed a surface oxide in coexistence with an oxygen adsorption phase on Ru(0001) when exposed to $10^{-4}$ mbar $O_2$ at 300–400 °C and connected the increase in activity to an expansion of the oxide phase. Therefore, it seems unlikely that $RuO_2$ or Ru surfaces stay in their fully oxidic or metallic state, respectively, when exposed to reaction conditions.

Overall, it can be summarized that the presence of multiple phases of $RuO_2$ and Ru has been linked to higher activity of the catalyst. Some of these phases may only be present under reaction conditions or in small fractions. As such, they could easily be missed, especially in non-operando measurements. This may also explain the controversial discussion in the literature about the active phase of $Ru/RuO_2$ in the CO oxidation. How multiple phases correlate to increased activity is, however, still unclear. According to the literature, it could be a core shell structure with a thin oxide layer on top of a metal core, a sub-stoichiometric $RuO_x$, or coexistence of these phases. In UHV studies, the reduction of $RuO_2$(110) and $RuO_2$(100) has shown to not lead to sub-stoichiometric $RuO_x$ phases.

Instead, the $RuO_2$ decomposes into $RuO_2$ and Ru (with adsorbed oxygen) patches under reducing reaction conditions [56].

Due to the dynamic behavior of the catalyst (cf. also discussion of Figure 6) depending on the applied gas composition, it is paramount to conduct operando spectroscopic experiments with supported powder catalysts. In this study, we investigated $RuO_2$ supported on rutile $TiO_2$ in a flow reactor cell adapted to a DRIFTS spectrometer. We found that $RuO_2@TiO_2$ and reduced $RuO_2@TiO_2$ revealed significant differences in activity when exposed to various CO oxidation reaction conditions (cf. Figures 3B and 5B). Regardless of the CO oxidation reaction conditions being reducing or oxidizing, reduced $RuO_2@TiO_2$ turned out to be always significantly more active than $RuO_2@TiO_2$.

Under oxidizing conditions, both samples ($RuO_2@TiO_2$ and reduced $RuO_2@TiO_2$) were similarly active at low temperatures, but at 100 °C the conversion on reduced-$RuO_2@TiO_2$ increased steeply and remained higher than that of $RuO_2@TiO_2$. It is important to note that the conversion for $RuO_2@TiO_2$ reached a plateau from 150 °C up to 180 °C and then increased again. Taking into consideration that a chemical reduction of $RuO_2@TiO_2$ can occur even under oxidizing reaction conditions (cf. Figure 4), the increased conversion above 180 °C is correlated with the reduction of $RuO_2@TiO_2$. This interpretation is supported by the fact that 180 °C is also the temperature at which reduction of $RuO_2@TiO_2$ under reducing reaction conditions sets in (cf. Figure 5A).

The activity behavior of the samples was quite similar under reducing conditions. Above 60 °C, reduced $RuO_2@TiO_2$ revealed higher conversions than $RuO_2@TiO_2$ throughout the entire temperature range. Although the conversion of $RuO_2@TiO_2$ did not exhibit a plateau, it only increased slightly in the temperature region of 160–200 °C. In this temperature region, a slight reduction of $RuO_2@TiO_2$ was observed in DRIFTS. For even higher temperatures, DRIFT spectra evidenced pronounced reduction concomitantly with an increase in CO conversion, thus again correlating chemical reduction of $RuO_2@TiO_2$ with an increase in activity.

Gao et al. [3] proposed that the interaction between Ru metal and $RuO_2$ may play an important role in CO oxidation on $RuO_2$. Farkas et al. [31,36] and Blume et al. [55] reported on a surface-phase separation into oxygen-rich and -depleted areas occurring on $RuO_2$ during CO oxidation, with the latter being the phase of enhanced activity. The increased activity of the partially reduced $RuO_2$ surface may even suggest bifunctionality such as that discussed for PdO. Weaver et al. [5] argued that on partially reduced PdO, CO oxidation is most favorable on the metal Pd surface, which is supplied with O from surrounding PdO. The bifunctionality of partially reduced $RuO_2$ would also be in line with the mechanistic arguments regarding too strong or too weak adsorption for CO and O on Ru and $RuO_2$. Boundary regions may offer adsorption sites with intermediary adsorption energies, which would be favorable for CO oxidation, according to the Sabatier principle.

The adsorbed CO species resulting in the 2040 $cm^{-1}$ band in DRIFTS seemed to be especially active. On the reduced catalyst, this species reacted off at ca. 100 °C under oxidizing conditions, leaving the high- and low-frequency bands in DRIFTS, which were consumed only above 200 °C (cf. Figure 3C). Under reducing conditions, the same species was observed in conjunction with an activity increase for $RuO_2@TiO_2$ at 200 °C; this CO species remained observable since excess CO was present in the gas phase. Under reducing conditions, the 2040 $cm^{-1}$ band was preserved up to 220 °C on reduced $RuO_2@TiO_2$ in conjunction with the sample showing significantly higher activity than its oxidized counterpart (cf. Figure 5A,C). In both cases ($RuO_2@TiO_2$ and reduced-$RuO_2@TiO_2$), the band was consumed at temperatures above 220 °C. The fact that the 2040 $cm^{-1}$ CO species was stable up to 200 °C under reducing conditions further corroborates that its disappearance at 100 °C under oxidizing conditions cannot have been due to desorption but actually was caused by a higher reactivity of this species. The reappearance of the 2040 $cm^{-1}$ band when the samples were cooled under reducing conditions (Figure S4) demonstrates that the corresponding sites remained present on the catalyst. A possible assignment of this band is that of CO adsorbed in the boundary regions between surface phases, most likely

metallic Ru (with adsorbed O) and $RuO_2$. This would also explain the absence of this band on partially oxidized $Ru^0 + TiO_2$ (cf. Figure 2B), as here no $RuO_2$ may be present.

An interesting observation about the low wavenumber band is its variable position (between 1985 $cm^{-1}$ and 2010 $cm^{-1}$) and asymmetric shape, whereas the position of the other bands in DRIFTS remained largely constant. This suggests that this band comprised CO species at various adsorption sites whose contributions changed depending on the chemical state of the catalyst. This interpretation would be in line with a varying composition of reduced $RuO_2$, as it is further reduced or re-oxidized. This behavior was especially apparent for reduced $RuO_2@TiO_2$ under oxidizing conditions between 120 and 180 °C. Here, the low wavenumber band shifted to higher wavenumbers and became more symmetric with higher reaction temperature (cf. Figure 3C) as the surface oxidized and approached the fully oxidized $RuO_2$.

Based on this discussion, we propose the following assignment of bands for partially reduced $RuO_2$ under CO oxidation conditions:

I. 1985–2030 $cm^{-1}$: CO on partially reduced $RuO_2$ with shape and position changing according to surface composition

II. 2040 $cm^{-1}$: CO sitting possibly in boundary region between surface phases with the highest activity.

III. 2075 $cm^{-1}$: CO on oxide $RuO_2$

IV. 2135 $cm^{-1}$: carbonyl $Ru^{x+}(CO)_y$, which can form on highly under-coordinated $Ru^{x+}$ sites. Reduced $RuO_2$ has been demonstrated to roughen [54,56], which would provide such under-coordinated $Ru^{x+}$ at the edges and corners [3].

Actually, very similar bands were reported by Gao et al. [3] when exposing Ru(0001) and $RuO_2$(110) to oxidizing and reducing reaction conditions, respectively, at 50 Torr and 500 K. For Ru(0001) under oxidizing conditions ($O_2$/CO = 5/1), at first no bands were observed; after 1–1.5 h bands at 2130, 2080, and 2040 $cm^{-1}$ appeared; and finally, after 2.5 h a single band at 2080 $cm^{-1}$ remained. This experiment is consistent with the surface going through partial oxidation and finally arriving at an oxide surface, according to our interpretation of the bands. Conversely, for $RuO_2$(110) under reducing conditions ($O_2$/CO = 1/10) the 2080 $cm^{-1}$ band was dominant at first. After 5 min of reduction the RAIR spectrum showed a weak low wavenumber shoulder and the 2130 $cm^{-1}$ band. During the next 6 h of reaction time, bands at 2050 and 2020 $cm^{-1}$ increased in intensity with the 2080 $cm^{-1}$ band diminishing. Altogether, these RAIRS data [3] on single-crystalline surfaces of Ru/$RuO_2$ are in reasonable agreement with our DRIFTS experiments of Ru-based powder catalysts supported on rutile $TiO_2$.

## 4. Experimental Details

### 4.1. Sample Preparation and Characterization

For DRIFTS experiments, we prepared $RuO_2$ supported on rutile $TiO_2$ ($RuO_2@TiO_2$) by a modified Pechini synthesis, as described in detail by Khalid et al. [57]. This ensured an even dispersion of the catalytically active oxides, which absorb IR radiation, on the reflective $TiO_2$ matrix. The BET surface area of the used support was 20 $m^2 \cdot g^{-1}$, with a mean particle size of 100 nm. Another advantage of this type of preparation is the better comparability of DRIFTS results with the kinetic data of previous work. The $RuO_2@TiO_2$ samples were thoroughly characterized by transmission electron microscopy (TEM), X-ray diffraction (XRD), and Raman spectroscopy in a previous study [57].

Two types of samples studied were oxidized and reduced samples of 2 mol% supported $RuO_2$. Two mol% $RuO_2$ on 20 $m^2$/g $TiO_2$ corresponds to an average thickness of 0.24 nm. Since $RuO_2$ is known to grow on $TiO_2$ with a thickness of no less than 3 monolayers or 1.5 nm [58], this corresponds to a surface coverage of maximally 16%. The samples obtained from the Pechini synthesis were first thermally oxidized or reduced by applying $O_2$ or CO at 300 °C. Although the supported $RuO_2$ could not be identified definitively by TEM micrographs, they showed no morphological differences between the oxidized and reduced samples (cf. Figure S5). These samples are referred to as $RuO_2@TiO_2$ and reduced

$RuO_2@TiO_2$, respectively, and are characterized by X-ray photoelectron spectroscopy (XPS). For XPS analysis, a PHI VersaProbe II instrument was used. The data were recorded employing a photon energy of 1486.6 eV (Al K$\alpha$ line).

DRIFT spectra of purely metallic Ru and purely oxidic $RuO_2$ samples were employed as references to assign the DRIFTS CO bands of $RuO_2@TiO_2$ under reaction conditions. Full reduction of our $RuO_2$ samples did not seem to be possible, as can be seen in Figure S1. To obtain the CO adsorption signals of pure metallic samples, we therefore used 33 w% of commercially available Ru (chempur) metal powder mixed with $TiO_2$. These samples were also reductively pretreated to remove any surface oxides that may have formed by applying 4% CO at 300 °C for 4 h. This sample is referred to as $Ru^0$ + $TiO_2$. After this treatment, XPS indicated pure metallic samples, as demonstrated in Figure 1.

Figure 1 also depicts XP spectra of $RuO_2@TiO_2$ and $Ru^0$ + $TiO_2$ after being exposed to a reducing reaction feed (4% CO/1% $O_2$/95% Ar) at 300 °C. Quite surprisingly, the spectra of $RuO_2@TiO_2$ and $Ru^0$ + $TiO_2$ exhibited both oxide and metal Ru signals. The samples were therefore in a state of partial reduction/oxidation.

*4.2. Reaction Conditions*

The reactor setup was built in house, and the description of it can be found in a recent publication by our group [10]. It consists of a gas supply controlled by mass flow controllers, a custom designed rector cell made from 1.4742 Ni-free steel, and a mass spectrometer for gas analysis. To derive the conversion from MS data, the $CO_2$ signal ($m/z = 44$) was normalized by its maximum value (at full conversion). Full consumption of $O_2$ (for reducing conditions) or CO (for oxidizing conditions) was used to determine the point at which full conversion was achieved. For oxidizing conditions, some CO always remained in the MS spectrum due to the cracking pattern of $CO_2$. In this case, full conversion was assumed when no further decrease in $m/z = 28$ and no further increase in $m/z = 44$ was observed with rising temperature.

Total flow for all experiments was 50 sccm, whereas the heating ramp was set as 1.8 K·min$^{-1}$. The gas compositions for the various experiments are compiled in Table 2. In order to minimize temperature changes induced by the reaction heat, by keeping the concentration of the reactants as small as possible, as well as due to limitations in the MFC flow range, the ratio of reactive gas to carrier gas had to be varied.

**Table 2.** Gas compositions for different experiments.

| Reaction Conditions | Ar/% | O$_2$/% | CO/% |
|:---:|:---:|:---:|:---:|
| Oxidizing | 96 | 2 | 2 |
| Reducing | 95 | 1 | 4 |
| CO only | 96 | - | 4 |

**5. Conclusions**

CO was employed as a probe molecule to study the actual oxidation state of a supported $RuO_2@TiO_2$ catalyst during the CO oxidation reaction. To do so, online mass spectrometry (MS) was coupled with operando diffuse reflectance infrared Fourier transform spectroscopy (DRIFTS). The CO DRIFT spectra of the pure oxide $RuO_2@TiO_2$ samples and the pure metal $Ru^0$ + $TiO_2$ were governed by a single mode at ca. 2060 cm$^{-1}$. Partially reduced $RuO_2@TiO_2$, on the other hand, was characterized by four distinct band regions in DRIFTS at 2135, 2075, 2040, and 2000 cm$^{-1}$. The combination of conversion (MS) and vibrational CO data (DRIFTS) revealed higher activity of reduced-$RuO_2@TiO_2$ than its oxidized counterpart both under reducing and oxidizing CO oxidation reaction conditions. The catalysts were shown to adopt the catalytically active phase dynamically to the reaction conditions independent of their initial state, only depending on the applied reaction mixture. Most surprisingly, even under oxidizing reaction conditions partial reduction of

$RuO_2@TiO_2$ was encountered. The CO species at 2040 cm$^{-1}$ in DRIFTS was shown to be especially reactive.

**Supplementary Materials:** The following supporting information can be downloaded at: https://www.mdpi.com/article/10.3390/catal13081178/s1, Table S1: XPS fit parameter; Figure S1: Ru 3d XP spectra of oxidized and reduced $RuO_2$; Figure S2: DRIFTS spectrum of CO adsorption on $Ru^0+TiO_2$, which has after oxidative pretreatment under 4% $O_2$ at 300 °C for 12h; Figure S3: IR spectra of pre-oxidized $TiO_2$ heated in a CO atmosphere; Figure S4: Operando DRIFT spectra of reduced-$RuO_2@TiO_2$ during cool down under reducing (1% $O_2$/4% CO/95% Ar) reaction feed composition; Figure S5: TEM micrographs of $RuO_2@TiO_2$ (left) before and (right) after reduction.

**Author Contributions:** Conceptualization, H.O. and P.T.; methodology, P.T.; XPS investigation, L.G. and T.W.; writing—original draft preparation, P.T. and H.O.; writing—review and editing, P.T., T.W., L.G. and H.O.; funding acquisition, H.O. All authors have read and agreed to the published version of the manuscript.

**Funding:** The research was funded by the Deutsche Forschungsgemeinschaft (DFG, German Research Foundation-493681475).

**Data Availability Statement:** The data used to support the findings of this study are included within the article.

**Acknowledgments:** P.T. and H.O. acknowledge financial support from the DFG (SPP2080—493681475).

**Conflicts of Interest:** The authors declare no conflict of interest.

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
