# Peer review of "Identifying the Active Phase of RuO2 in the Catalytic CO Oxidation Reaction, Employing Operando CO Infrared Spectroscopy and Online Mass Spectrometry"

_catalysts, doi:10.3390/catal13081178_

Round 1
Reviewer 1 Report
The authors present an operando study correlating DRIFTS and catalytic activity measurements for ruthenium catalysts. The paper is an interesting and useful contribution to the ongoing discussion of the active phase for CO oxidation. The data demonstrate that the catalytst is in a mixed oxidation state under reaction conditions, and identify a CO adsorption state that appears correlated with the low-temperature activity. My feeling is that the extent to which this study "resolve[s] the long-standing debate concerning the active phase" (abstract) is somewhat overstated; interpretation of the catalyst structure in terms of IR bands remains rather ambiguous I would say, and a specific active site and mechanism is not identified. The results are interesting nevertheless and I would support publication following some revision.
Specific comments/requests:
-In the abstract, revise the sentence "to resolve the long-standing debate" in order to avoid implying that the debate is hereby resolved.
-In the introduction it is stated that understanding the CO oxidation reaction will help in the study of other processes which are somewhat more relevant for technology. This is rather vague at the moment; how specifically is this related to the Deacon process, CER, OER, etc?
-The description of the metalic Ru catalysts is quite confusing. You refer to "Ru0+TiO2" in the text but the actual material studied was just metal powder? In this case there should be nothing about TiO2 in the name. Please apply consistent and sensible nomenclature to help the reader follow the discussion.
-The fact that the conversion is calculated from the CO2 partial pressure assuming full conversion in the limit of high temperature is stated repeatedly. The description in the methods section is sufficient, delete this elsewhere. Have you checked that e.g. varying the flow rate does not affect the CO2 concentration at high temperature?
-What is the surface area of your TiO2 support? What does a 2% loading of RuO2 correspond to in terms of average thickness?
-The discussion includes a paragraph (p11) commenting on the limitations of batch reactor studies, with the implication that these are not relevant here where flow conditions are used. But under flow conditions at high conversions, the same effects (mass transfer limitations and the drastically changing gas composition) impact the interpretation. What are you trying to convey with this paragraph?
-Bands in the DRIFT spectra become negative at high temperatures. Why is this? does this affect the adsorbed CO states in addition to the gas phase and higher wavenumber features?
-I am a bit confused about the identification of the different CO stretching bands, summarized in Table 2. When you write "Bridging" and "On-top" are you referring to sites on RuO2(110)? or on Ru metal? Which bands represent adsorption on the metal and which on the oxide? Somehow this should be clarified, also in the related discussion.
-Fig. 1: the fitted XPS peaks seem not to have consistent lineshapes; the difference between the Ru4+ peaks in B and D stands out especially. Is this justified? Can you give a bit more detail about the fit procedures used here?
English is good overall, but there are a fairly large number of small typos; please go through careful grammar and spellchecking.
The language is somewhat informal in places; "never-ending discussions", "not that fancy". Please revise this.
Reviewer 2 Report
In this manuscript, the operando study using DRIFTS and on-line MS was applied for CO oxidation over Ru-based catalysts. The results and discussion delivered in this manuscript is so fundamental and intuitive that they will be helpful to lots of readers once published. Therefore, I recommend this manuscript for publication in this journal. Here are some minor comments:
1. There are many weird expressions along the manuscript: “Error! Reference source not found”. Due to this problem, it is difficult to follow the results and discussion smoothly.
2. The stated wavenumbers in the text should be identical to those in corresponding figures.
3. The authors are careful for the subscript notation.
4. The y-axis of many figures has major/minor ticks while the axis title and unit are not provided. This should be revised.
The English usage and grammar can be improved in several sentences.
Reviewer 3 Report
The author of this manuscript studied the loading type by combining online mass spectrometry (MS) with diffuse reflectance infrared Fourier transform spectroscopy (DRIFTS) using CO as a probe molecule for RuO2@TiO2 on the actual oxidation state and reaction activity of the catalyst in CO oxidation reaction. The results indicate that the dynamic adaptation of the catalyst to reaction conditions is not related to the initial state of the catalyst, but only depends on the reaction mixture applied. Under the conditions of reduction and oxidation of CO oxidation reaction, reduced-RuO2@TiO2 Their activity is higher than that of oxidation products. However, I believe that there are many issues that need to be addressed in research before being accepted by any journal.
1: In Fig. 2, both Ru0+TiO2 and RuO2@TiO2 samples result in incomplete oxidation or reduction after oxidation and reduction post-treatment, respectively, so please explain why the number, location, and intensity of the DRIFT wave peaks are completely different between the two samples?
2: The wave number at 2000cm-1 in Fig. 3B shifts towards higher wave numbers as the temperature increases, it does not remain constant. Please explain the reason.
3: The annotation on Fig. 5C is 2134cm-1, but the language description was 2135cm-1. Pay attention to the consistency of the image and text.
4: There is no picture of the morphology of the samples in the paper, did the morphology of the samples change during oxidation or reduction? Could the catalytic performance be related to the sample morphology and defects? Please add the SEM images of pure RuO2@TiO2, Ru0+TiO2 and reduced-RuO2@TiO2.
5: A large number of 'Error! Reference source not found.' appear in the article, which has seriously affected normal reading and understanding. Pay attention to the writing format of the article and make timely corrections.
